# Probabilistic Analysis of Composite Materials with Hyper-Elastic Components

**DOI:** 10.3390/ma15248878

**Published:** 2022-12-12

**Authors:** Marcin Kamiński, Damian Sokołowski

**Affiliations:** Department of Structural Mechanics, Faculty of Civil Engineering, Architecture & Environmental Engineering, Łódź University of Technology, Al. Politechniki 6, 90-924 Lodz, Poland

**Keywords:** composites, hyper-elasticity, homogenization, probabilistic methods, interface defects, rubber-like materials

## Abstract

This work is a comprehensive literature overview in the area of probabilistic methods related to composite materials with components exhibiting hyper-elastic constitutive behavior. A practical area of potential applications is seen to be rubber, rubber-like, or even rubber-based heterogeneous media, which have a huge importance in civil, mechanical, environmental, and aerospace engineering. The overview proposed and related discussion starts with some general introductory remarks and a general overview of the theories and methods of hyper-elastic material with a special emphasis on the recent progress. Further, a detailed review of the current trends in probabilistic methods is provided, which is followed by a literature perspective on the theoretical, experimental, and numerical treatments of interphase composites. The most important part of this work is a discussion of the up-to-date methods and works that used the homogenization method and effective medium analysis. There is a specific focus on random composites with and without any interface defects, but the approaches recalled here may also serve as well in sensitivity analysis and optimization studies. This discussion may be especially helpful in all engineering analyses and models related to the reliability of elastomers, whose applicability range, which includes energy absorbers, automotive details, sportswear, and the elements of water supply networks, is still increasing, as well as areas where a stochastic response is the basis of some limit functions that are fundamental for such composites in structural health monitoring.

## 1. Introductory Remarks

The computational analysis of materials is now standard and decisive for the design of contemporary appliances, mechanical parts, and bearing systems that are made of both homogeneous and heterogeneous (composite) materials. It is especially remarkable in the area of polymeric materials [1] that are reinforced or filled with some other specific micro-injections or nano-particles [2,3,4,5]. Such materials modeling needs a specific approach that is most frequently based upon macro-modeling that is linked with a calculation of the overall properties [6]; nevertheless, it may demand precise experimentally verified knowledge concerning the nano-mechanics of the reinforcements [7]. There is no doubt that numerical simulation changed the entire design process by supplementing the traditional cycle of conceptualizing and laboratory testing with various modeling tools, including sensitivity analyses [8], stochastic models [9] that include various nonlinearities [10], accounting for the anisotropy of fillers [11], atomistic modeling [12], and multiscale approaches [13], finally leading to reliability assessments [14]. It cuts down the design time and provides tools for a very optimized or complex solution that is unavailable when using even a very advanced analytical approach. All computations are based on constitutive models of materials, which involves defining their behavior under different mechanical, thermal [15], electrical [16], magnetic, and coupled [17] conditions. They are well-defined for relatively simple continua [18,19,20] but are still challenging for composites outside of the elastic region. In this review, the hyper-elastic response of a complex continuum is studied. It is highly useful for predicting the behavior of rubber-like composites with a polymeric matrix. Their main strengths include their ease of usage and calibration, computational efficiency, and flexibility of usage and accessibility in commercial codes. They could also be quite easily augmented to capture hysteresis in cyclic loading. Applications of various hyper-elastic constitutive models range from the tire industry to biological tissues, such as human arteries [21] and polymers, and include civil engineering, where such materials are applied as vibration dampers and protection.

This review work consists of five sections, starting with a retrospective look into hyper-elastic materials and their constitutive models. Then, a relatively short description and literature overview are given related to the probabilistic methods that are available in modern engineering. The next part is devoted to the characterization of interphases and interface defects that are inherent in composite materials. Further, multiscale models of composite materials and the additional numerical simulations are briefly characterized, together with their recent advances. The key milestones and very recent ideas in the homogenization method end the entire review.

## 2. Hyper-Elastic Materials

An early motivation for the theoretical formulation of hyper-elastic materials was the lack of existence of large-strain models capturing deformations that accounted for their non-linearity above an infinitesimal level. Such materials can deform significantly and nonlinearly upon loading without breaking and then return to their initial configuration. Such rubber elasticity is achieved due to very flexible long-chain molecules and a three-dimensional network structure that is formed via cross-linking or some entanglements between molecules. This behavior characterizes a wide range of continua, including rubber-like materials, polymers [22], and elastomers [23]. Its implementation in the finite element framework requires two important ingredients to solve the given boundary value problem. They are the stress tensor and the consistent fourth-order tangent operator; the latter is the result of linearization of the former rubber-like materials, which are generally modeled as homogeneous, isotropic, incompressible or nearly incompressible, geometrically and physically nonlinear, hyper- or visco-elastic solids [24,25,26,27,28] and visco-plastic solids [29]. Their models are commonly supported by experimental data. The most common tests involve uniaxial tension, biaxial tension, and pure shear. Some models also consider aging [30,31], the Mullins effect [32,33,34], hysteresis [35], or the failure of rubber-like materials [36,37]. Hyper-elastic models are reviewed in this work and their theoretical introduction is available, for example, in [38,39,40,41], whereas some numerical illustrations are contained in [42,43,44,45]. Let us recall the basic concepts for hyper-elastic constitutive models, which can be generally divided into three essentially different categories: phenomenological models [46,47], micromechanical approaches, and constitutive theories obtained with the use of the artificial neural networks (ANNs [48,49,50,51,52]). They are all presented graphically in Figure 1, Figure 2, Figure 3 and Figure 4, respectively.

Phenomenological models are generally based on invariants, the stretch ratio, or both. Their parameters do not bear any physical interpretation. Invariant-based phenomenological models can be further divided into models that leverage a series function, including the limitation of chain extensibility, or are based on a logarithmic function, power law, or exponential function. They were developed in the mid- and late-twentieth century and provide all analytical solutions for an isotropic single-phase medium. The number of parameters included in their equations is small, which enables their analytical treatment of simple engineering problems. Contemporary models introduce more parameters that increase the flexibility and accuracy of calculations. The trade-off is a considerable increase in calibration and computation efforts that makes them preferable for large-scale computer simulations and not for analytical calculus.

Micromechanical models are based on the analysis of networks of cross-linked long-chain molecules. They could be subdivided into these based on a Gaussian or non-Gaussian network model, or involve a mixture of the two. Artificial neural networks could be classical and bear no mechanical constraints, or constitutive and include such constraints in the network. A comparison of various constitutive models has been provided below and notation applied in the relevant formulas is the following one: W (with various upper superscripts) denotes a function of deformation energy density, σ_11_ denotes the true Cauchy stress under uniaxial tension, μ is the shear modulus (the second Lame constant), ν is the Poisson ratio, λ_i_ denote principal stretches of the material (λ in uniaxial state) and J is their product. Moreover, I_1_ is the first invariant of the right Cauchy-Green deformation tensor, **F** is the deformation gradient, λ_L_ is the first Lame constant, while C_i_ and D*i* as well as A_p_, B_p_, α_p_ and β_p_ are different material constants in the constitutive theories presented below. The Readers are encouraged to look into the given references to find a specific physical or mathematical explanation of some other parameters appearing below. 

The first proposed hyper-elastic models were invariant-based, specifically the series function ones. The first theoretical approach was done by Mooney and Rivlin [53,54]. The model includes the first two invariants in their first powers:(1)$$W^{\mathrm{MR}}{=C}_{1}\left(I_{1}-3\right){+C}_{2}\left(I_{2}-3\right)$$
(2)$$\sigma_{11}^{\mathrm{MR}}=\left(2C_{1,\mathrm{MR}}^{m}+\frac{2C_{2,\mathrm{MR}}^{m}}{\lambda }\right)\left(\lambda^{2}-\frac{1}{\lambda }\right)$$


This model was simplified to include only the first invariant; it is widely known as the neo-Hookean model [55,56,57]:(3)$$W^{\mathrm{NH}}{=C}_{1}\left(I_{1}-3\right)$$
(4)$$\sigma_{11}^{\mathrm{NH}}{=2C}_{1}\left(\lambda -\frac{1}{\lambda^{2}}\right)$$

A third, second-power term was included by Isihara [58]:(5)$$W^{I}{=C}_{10}\left(I_{1}-3\right){+C}_{20}{\left(I_{1}-3\right)}^{2}{+C}_{01}\left(I_{2}-3\right)$$
(6)$$\sigma_{11}^{I}{=C}_{10}\left(\lambda -\frac{1}{\lambda^{2}}\right)+\left(\lambda^{3}+\frac{C_{20}}{C_{01}}-\left[1+\frac{C_{20}}{C_{01}}\right]\frac{1}{\lambda^{3}}\right)$$

Next, a third term for the first invariant was added by Yeoh [59]:(7)$$W^{Y}{=C}_{1}\left(I_{1}-3\right){+C}_{2}{\left(I_{1}-3\right)}^{2}$$
(8)$$\sigma_{11}^{Y}=2\left(C_{1}{+2C}_{2}\left(I_{1}-3\right){+3C}_{3}{\left(I_{1}-3\right)}^{2}\right)\left(\lambda -\frac{1}{\lambda^{2}}\right)$$

Carroll’s model drops the conventional $\left(I_{i}-3\right)$ form [60] to give
(9)$$W^{C}{=\beta }_{1}I_{1}{+\beta }_{2}I_{1}^{4}{+\beta }_{3}\sqrt{I_{2}}$$
(10)$$\sigma_{11}^{C}=\left[{2\beta }_{1}{+8\beta }_{2}{\left(\frac{2}{\lambda }{+\lambda }^{2}\right)}^{3}{+\beta }_{3}{\left({1+\lambda }^{3}\right)}^{-1/2}\right]\left[\lambda -\frac{1}{\lambda^{2}}\right]$$
It violates the restriction introduced by Ogden and Treolar that $U\left(I_{1}{,I}_{2}{,I}_{3}\right)=0$ in the reference configuration and the following modification was proposed to overcome this peculiarity:(11)$$W^{\mathrm{CM}}{=\beta }_{1}\left(I_{1}-3\right){+\beta }_{2}\left(I_{1}^{4}-81\right){+\beta }_{3}\left(\sqrt{I_{2}}-\sqrt{3}\right)$$
(12)$$\sigma_{11}^{\mathrm{CM}}=\left[{2\beta }_{1}{+8\beta }_{2}{\left(\frac{2}{\lambda }{+\lambda }^{2}\right)}^{3}{+\beta }_{3}{\left({1+\lambda }^{3}\right)}^{-1/2}\right]\left[\lambda -\frac{1}{\lambda^{2}}\right]$$

It is called a modified Carrol’s model and is presented here in the incompressible form [61]. The last series function model was introduced by Zhao [62]. It adds terms with mixed invariants in different powers governed by the same constant. This concept was also used before by Bahreman and Darijani.
(13)$$W^{Z}=C_{-1}^{1}\left(I_{2}-3\right)+C_{1}^{1}\left(I_{1}-3\right)+C_{2}^{1}\left(I_{1}^{2}{-2I}_{2}-3\right)+C_{2}^{1}{\left(I_{1}^{2}{-2I}_{2}-3\right)}^{2}$$
(14)$$\sigma_{11}^{Z}=\left[{2\beta }_{1}{+8\beta }_{2}{\left(\frac{2}{\lambda }{+\lambda }^{2}\right)}^{3}{+\beta }_{3}{\left({1+\lambda }^{3}\right)}^{-1/2}\right]\left[\lambda \;-\frac{1}{\lambda^{2}}\right]$$

The second most common variants of phenomenological models are power, exponential, or logarithmic ones. The first one was proposed by Knowles [63]:(15)$$W^{K}=\frac{\mu }{2b}\left({\left(1+\frac{b\left(I_{1}-3\right)}{n}\right)}^{n}-1\right)$$
(16)$$\sigma_{11}^{S}=\left(\frac{1}{2}\overset{N}{\underset{p=1}{\sum }}A_{p}{\left(\frac{I_{1}}{3}\right)}^{\alpha_{p}}+\frac{1}{2}\overset{N}{\underset{p=1}{\sum }}B_{p}{\left(\frac{I_{1}}{3}\right)}^{\beta_{p}}\right)\left(\lambda \;-\frac{1}{\lambda^{2}}\right)$$

The next one was proposed by Swanson [64]:(17)$$W^{S}=\frac{3}{2}\overset{N}{\underset{p=1}{\sum }}\frac{A_{p}}{{1+\alpha }_{p}}{\left(\frac{I_{1}}{3}\right)}^{{1+\alpha }_{p}}+\frac{3}{2}\overset{N}{\underset{p=1}{\sum }}\frac{B_{p}}{{1+\beta }_{p}}{\left(\frac{I_{1}}{3}\right)}^{{1+\beta }_{p}}$$
(18)$$\sigma_{11}^{S}=\left(\frac{1}{2}\overset{N}{\underset{p=1}{\sum }}A_{p}{\left(\frac{I_{1}}{3}\right)}^{\alpha_{p}}+\frac{1}{2}\overset{N}{\underset{p=1}{\sum }}B_{p}{\left(\frac{I_{1}}{3}\right)}^{\beta_{p}}\right)\left(\lambda \;-\frac{1}{\lambda^{2}}\right)$$

The next model was proposed by Gregory [65]:(19)$$W^{G}=\frac{A}{2\left(1-n/2\right)}{\left(I_{1}{-3+C}^{2}\right)}^{1-n/2}+\frac{B}{2\left(1-m/2\right)}{\left(I_{1}{-3+C}^{2}\right)}^{1-m/2}$$
(20)$$\sigma_{11}^{G}=\left[A{\left(I_{1}{\;-\;3+C}^{2}\right)}^{n/2}+B{\left(I_{1}{-3+N}^{2}\right)}^{m/2}\right]\left[{\lambda -\lambda }^{-2}\right]$$

A further modification was invented recently by Hong et al. and it was called a modified Gregory model [47]:(21)$$W^{\mathrm{MG}}=\frac{A}{1+\alpha }{\left(I_{1}{-3+M}^{2}\right)}^{1+\alpha }+\frac{B}{1+\beta }{\left(I_{1}{-3+N}^{2}\right)}^{1+\beta }$$
(22)$$\sigma_{11}^{\mathrm{MG}}=2\left[A{\left(I_{1}{-3+M}^{2}\right)}^{\alpha }+B{\left(I_{1}{-3+N}^{2}\right)}^{1+\beta }\right]\left[{\lambda -\lambda }^{-2}\right]$$

Another variant of a power-law-based function was put forward by Lopez-Pamies [66], where
(23)$$W^{\mathrm{LP}}=\overset{M}{\underset{r=1}{\sum }}\frac{3^{{1-\alpha }_{r}}}{{2\alpha }_{r}}\mu_{r}\left[{\bar{I}}_{1}^{\alpha_{r}}{-3}^{\alpha_{r}}\right]$$
(24)$$\sigma_{11}^{\mathrm{LP}}=\frac{\lambda^{3}-1}{{2\lambda +\lambda }^{4}}\overset{M}{\underset{r=1}{\sum }}3^{{1-\alpha }_{r}}\mu_{r}{\left(\lambda^{2}+\frac{2}{\lambda }\right)}^{\alpha_{r}}$$

This type of function was also proposed by Yeoh in his modified form by adding an exponential term [67]:(25)$$W^{Y}=C_{1}\left(I_{1}-3\right)+C_{2}{\left(I_{1}-3\right)}^{2}+C_{3}{\left(I_{1}-3\right)}^{3}+\frac{\alpha }{\beta }\left({1-e}^{-\beta \left(I_{1}-3\right)}\right)$$
(26)$$\sigma_{11}^{Y}=2\left(C_{1}{+2C}_{2}\left(I_{1}-3\right){+3C}_{3}{\left(I_{1}-3\right)}^{2}\right)\left(\lambda -\frac{1}{\lambda^{2}}\right)$$

This was generalized in 2019 by Travis et al. to include arbitrary powers of all three terms. It was further called the generalized Yeoh model [68].
(27)$$W^{\mathrm{GY}}{=C}_{1}{\left(I_{1}-3\right)}^{m}{+C}_{2}{\left(I_{1}-3\right)}^{p}$$
(28)$$\sigma_{11}^{\mathrm{GY}}=2\left(\lambda -\frac{1}{\lambda^{2}}\right){\mathrm{mC}}_{1}{\left(I_{1}-3\right)}^{m-1}{+\mathrm{pC}}_{2}{\left(I_{1}-3\right)}^{p-1}{+\mathrm{qC}}_{3}{\left(I_{1}-3\right)}^{q-1}$$

Gent and Thomas moved away from using a power function and instead proposed a linear term combined with a relatively simple logarithmic term that allowed for a relatively easy stress formulation [69]:(29)$$W^{\mathrm{GT}}{=C}_{1}\left(I_{1}-3\right){+C}_{2}\ln\left[\frac{I_{2}}{3}\right]$$
(30)$$\sigma_{11}^{\mathrm{GT}}=\left(C_{1}+\frac{C_{2}}{\lambda }\right)\left(\lambda -\frac{1}{\lambda^{2}}\right)$$

These logarithmic or (later) exponential terms are incorporated into their energy functions to allow for a better estimation of deformation in uniaxial tension and equi-biaxial tension.

The first proposal of a combined exponential and logarithmic function was put forward by Hart-Smith [70]:(31)$$W^{\mathrm{HS}}{=c}_{10}\int \exp\left(c_{1}{\left[{\bar{I}}_{1}-3\right]}^{2}\right)d{\bar{I}}_{1}{+c}_{01}\ln\left(\frac{{\bar{I}}_{2}}{3}\right)$$
(32)$$\sigma_{11}^{\mathrm{HS}}=2\left[c_{10}\exp\left(c_{1}{\left[I_{1}-3\right]}^{2}\right)+\frac{c_{01}}{{\lambda I}_{2}}\right]\left[\lambda -\frac{1}{\lambda^{2}}\right]$$

Another variant of such a combination was proposed by Alexander [71]:(33)$$W^{A}{=c}_{1}\int \exp\left(c_{3}{\left[\bar{I_{1}}-3\right]}^{2}\right)d\bar{I_{1}}{+c}_{2}\ln\left(\frac{\bar{I_{2}}{-3+c}_{4}}{c_{4}}\right){+c}_{3}\left[\bar{I_{1}}-3\right]$$
(34)$$\sigma_{11}^{A}=2\left(c_{1}\exp\left(c_{3}{\left[I_{1}-3\right]}^{2}\right)+\frac{1}{\lambda }\left[\frac{c_{2}c_{4}}{I_{2}{-3+c}_{4}}{+c}_{3}\right]\right)\left({\lambda -\lambda }^{-2}\right)$$

Some other variations were put forward by Veronda and Westmann, Vito, Humphrey and Yin, and much later by Mansouri and Darijani.

Another combination of such a phenomenological model was introduced by Hoss and Marczak [72]. It consists of three terms, namely, power, exponential, and logarithmic terms:(35)$$W^{\mathrm{HM}}=\frac{\alpha }{\beta }\left({1-e}^{-\beta \left(I_{1}-3\right)}\right)+\frac{\mu }{2b}\left({\left(1+\frac{b\left(I_{1}-3\right)}{n}\right)}^{n}-1\right){+C}_{2}\ln\left(\frac{1}{3}I_{2}\right)$$
(36)$$\sigma_{11}^{\mathrm{HM}}=2\left(\lambda -\frac{1}{\lambda^{2}}\right)\left({\alpha e}^{-\beta \left(I_{1}-3\right)}+\frac{\mu }{2}{\left(1+\frac{b\left(I_{1}-3\right)}{n}\right)}^{n-1}+\frac{1}{\lambda }\frac{C_{2}}{I_{2}}\right)$$

Next is the exponential-linear model [73], which instead of the summation of the logarithmic and linear terms, proposed their multiplication:(37)$$W^{\mathrm{EL}}=A\left(\frac{1}{\alpha }e^{a\left(I_{1}-3\right)}+b\left(I_{1}-2\right)\left(1-\ln\left(I_{1}-2\right)\right)-\frac{1}{a}-b\right)$$
(38)$$\sigma_{11}^{\mathrm{EL}}=\frac{\lambda^{3}-1}{{2\lambda +\lambda }^{4}}\overset{M}{\underset{r=1}{\sum }}3^{{1-\alpha }_{r}}\mu_{r}{\left(\lambda^{2}+\frac{2}{\lambda }\right)}^{\alpha_{r}}$$

The most recent form of this type of model is the Anssari–Benam–Bucchi model [74]:(39)$$W^{\mathrm{ABB}}=\mu N\left(\frac{1}{6N}\left(I_{1}-3\right)-\ln\left(\frac{I_{1}-3N}{3-3N}\right)\right)$$
(40)$$\sigma_{11}^{\mathrm{ABB}}=2\mu \frac{9N-\left(\lambda^{2}{+2\lambda }^{-1}\right)}{3N-\left(\lambda^{2}{+2\lambda }^{-1}\right)}\left(\lambda^{2}-\frac{1}{\lambda }\right)$$

Another type of phenomenological model limits chain extensibility. This type introduces a limit for the stretch ratio for the macromolecular chain of rubber $I_{m}$ and provides other parameters for small strains. The first one was introduced by Warner [75]:(41)$$W^{W}=-\frac{\mu I_{m}}{2}ln\left(1-\frac{I_{1}-3}{I_{m}-3}\right)$$
(42)$$\sigma_{11}^{W}=\left(\frac{1}{2}\overset{N}{\underset{p=1}{\sum }}A_{p}{\left(\frac{I_{1}}{3}\right)}^{\alpha_{p}}+\frac{1}{2}\overset{N}{\underset{p=1}{\sum }}B_{p}{\left(\frac{I_{1}}{3}\right)}^{\beta_{p}}\right)\left(\lambda -\frac{1}{\lambda^{2}}\right)$$

A further proposal with a much more elaborate logarithmic term was made by van der Waals [76]. It includes the limit stretch $\lambda_{m}$:(43)$$W^{\mathrm{VW}}=-\mu \left[\lambda_{m}-3\right]\left[\ln\left(1-\theta \right)+\theta \right]-\frac{2}{3}{\left(\frac{\tilde{I}-3}{2}\right)}^{\frac{3}{2}},\;\theta =\sqrt{\frac{\hat{I}-3}{\lambda_{m}^{2}-3}\;},\hat{I}{=\beta I}_{1}+\left(1-\beta \right)I_{2}$$
(44)$$\sigma_{11}^{\mathrm{VW}}=\left(\left[{\beta +\lambda }^{-1}{-\beta \lambda }^{-1}\right]\left[\frac{\mu }{1-\eta }-a\mu {\left(\frac{\tilde{I}-3}{2}\right)}^{1/2}\right]\right)\left({\lambda -\lambda }^{-2}\right)$$

A small difference in the quotient was then proposed by Gent [77]. It allowed for better accuracy for small stretches:(45)$$W^{G}=-\frac{{\mu J}_{m}}{2}\ln\left[1-\frac{I_{1}-3}{J_{m}}\right]$$
(46)$$\sigma_{11}^{G}=\left(\frac{{\mu J}_{m}}{J_{m}{-I}_{1}+3}\right)\left(\lambda -\frac{1}{\lambda^{2}}\right),\;J_{m}{=I}_{m}{-3,\;I}_{1}{=\lambda }^{2}+\frac{2}{\lambda }$$

In the early 2000s, Pucci and Saccomandi [78] proposed a model with two logarithmic terms:(47)$$W^{\mathrm{PS}}{=-J}_{m}\frac{\mu }{2}\ln\left(1-\frac{\bar{I_{1}}-3}{J_{m}}\right){+c}_{2}\ln\left(\frac{\bar{I_{2}}}{3}\right)$$
(48)$$\sigma_{11}^{\mathrm{PS}}=\left(\frac{J_{m}\mu }{J_{m}{-I}_{1}+3}+\frac{{2c}_{2}}{{\lambda I}_{2}}\right)\left({\lambda -\lambda }^{-2}\right)$$

The last variant of this kind of phenomenological model was put forward by Horgan and Murphy [79]:(49)$$W^{\mathrm{HM}}=\frac{2\mu \left(I_{m}-3\right)}{c^{2}}\ln\left(1-\frac{\lambda_{1}^{c}{+\lambda }_{2}^{c}{+\lambda }_{3}^{c}}{I_{m}-3}\right)$$
(50)$$\sigma_{11}^{\mathrm{HS}}=\left(\frac{J_{m}\mu }{J_{m}{-I}_{1}+3}+\frac{{2c}_{2}}{{\lambda I}_{2}}\right)\left({\lambda -\lambda }^{-2}\right)$$

Phenomenological models based on the stretch ratio are simple, yet could be used for large strain ratios, especially the famous Ogden variant. The first one was proposed by Valanis and Landel [80]:(51)$$W^{\mathrm{VL}}=2\mu \sum_{i=1}^{3}\left(\lambda_{i}\ln\left(\lambda_{i}-1\right)\right)$$
(52)$$\sigma_{11}^{\mathrm{VL}}{=I}_{3}^{-1/2}\lambda_{i}\frac{\partial W}{\partial \lambda_{i}}$$

Its compressible form and additional proofs and its experimental validation is also provided in [81]. Further proposals were prepared by Ogden [14]:(53)$$W^{O}=\overset{M}{\underset{r=1}{\sum }}\frac{\mu_{r}}{\alpha_{r}}\left[\lambda_{1}^{\alpha_{r}}{+\lambda }_{2}^{\alpha_{r}}{+\lambda }_{3}^{\alpha_{r}}-3\right]$$
(54)$$\sigma_{11}^{O}=2\frac{\mu_{r}}{\alpha_{r}}\lambda^{\alpha_{r}}-{\frac{1}{\sqrt{\lambda }}}^{\alpha_{r}}$$

Shariff [13]:(55)$$W^{S}=\omega \left(\lambda_{1}\right)+\omega \left(\lambda_{2}\right)+\omega \left(\lambda_{3}\right)=E\overset{n}{\underset{j=0}{\sum }}\alpha_{j}\varphi_{j}\left(\lambda_{i}\right)$$
(56)$$\sigma_{11}^{S}=\frac{E}{\lambda }\left[\begin{array}{c} {\ln\lambda +\alpha }_{1}\vartheta_{1}{+\alpha }_{2}\vartheta_{1}\\ {+\alpha }_{3}\left[\frac{\vartheta_{2}}{\lambda^{3.6}}-\frac{\vartheta_{3}}{\lambda^{-1.8}}\right]\\ {+\alpha }_{4}\left[\vartheta_{2}-\vartheta_{3}\right] \end{array}\right]\;\vartheta_{1}=\left[e^{1-\lambda }{-e}^{{1-\lambda }^{\frac{1}{2}}}{+\lambda -\lambda }^{-\frac{1}{2}}\right],\;\vartheta_{2}={\left(\lambda -1\right)}^{3},\;\vartheta_{3}={\left(\lambda^{-\frac{1}{2}}-1\right)}^{3}$$Attard and Hunt [25]:(57)$$W^{\mathrm{AH}}=\sum_{r=1}^{M}\frac{A_{r}}{2r}\mathrm{tr}\left({\bar{C}}^{r}-I\right)+\frac{B_{r}}{2r}\mathrm{tr}\left({\bar{C}}^{r}-I\right),\;\mathrm{tr}\left({\bar{C}}^{r}-I\right)=\left[{\bar{\lambda }}_{1}{}^{2r}+{\bar{\lambda }}_{2}{}^{2r}+{\bar{\lambda }}_{3}{}^{2r}-3\right]$$
(58)$$\sigma_{11}^{\mathrm{AH}}=\overset{M}{\underset{r=1}{\sum }}A_{r}\left[\lambda^{2r-1}{-\lambda }^{-r-1}\right]{+B}_{r}\left[\lambda^{r-1}{-\lambda -}^{2r-1}\right]$$and Arman and Narooei [82]:(59)$$W^{\mathrm{AN}}=\overset{N}{\underset{p=1}{\sum }}A_{p}\left(exp\left[m_{p}\left(\lambda_{1}^{\alpha_{p}}+\lambda_{2}^{\alpha_{p}}+\lambda_{3}^{\alpha_{p}}-3\right)\right]-1\right)+\overset{N}{\underset{q=1}{\sum }}B_{q}\left(exp\left[n_{q}\left(\lambda_{1}^{-\beta_{q}}+\lambda_{2}^{-\beta_{q}}+\lambda_{3}^{-\beta_{q}}-3\right)\right]-1\right)$$
(60)$$\sigma_{11}^{\mathrm{AN}}=\frac{E}{\lambda }\left[\begin{array}{c} {\ln\lambda +\alpha }_{1}\vartheta_{1}{+\alpha }_{2}\vartheta_{1}\\ {+\alpha }_{3}\left[\frac{\vartheta_{2}}{\lambda^{3.6}}-\frac{\vartheta_{3}}{\lambda^{-1.8}}\right]\\ {+\alpha }_{4}\left[\vartheta_{2}-\vartheta_{3}\right] \end{array}\right]$$

Mixed phenomenological models use invariants and stretch ratios that can capture small and high strains well. They emerged in the early 2000s. Some examples are the continuum hybrid model [83]:(61)$$W^{\mathrm{CH}}{=K}_{1}\left(I_{1}-3\right){+K}_{2}\ln\frac{I_{2}}{3}+\frac{\mu }{\alpha }\left(\lambda_{1}^{\alpha }{+\lambda }_{2}^{\alpha }{+\lambda }_{3}^{\alpha }-3\right)$$
(62)$$\sigma_{11}^{\mathrm{CH}}{=I}_{3}^{-1/2}\lambda_{i}\frac{\partial U}{\partial \lambda_{i}}$$
and the WFB model [84]:(63)$$W^{\mathrm{WFB}}=\int_{1}^{L_{f}}\left(F\left(\lambda_{1}\right)A\left(\lambda_{1}e^{{-\mathrm{BI}}_{1}}\right)+C\left(\lambda_{1}I_{1}^{-D}\right)\right)\left(\lambda_{1}^{2}-\frac{1}{\lambda_{1}}\right){d\lambda }_{1}$$
(64)$$\sigma_{11}^{\mathrm{WFB}}{=I}_{3}^{-1/2}\lambda_{i}\frac{\partial U}{\partial \lambda_{i}}$$

Micromechanical network models aim to determine the microstructural mechanisms of the material that relate to their mechanical properties. They can be categorized by the presence of a Gaussian character in the chain network. Gaussian chain network models characterize small and moderate stretches well but are not as accurate for large ones with hardening. The first Gaussian model was proposed by Treloar [85]:(65)$$W^{G}=\frac{1}{2}NkT\left(\lambda_{1}^{2}{+\lambda }_{2}^{2}{+\lambda }_{3}^{2}-3\right)$$
(66)$$\sigma_{11}^{G}{=I}_{3}^{-1/2}\lambda_{i}\frac{\partial U}{\partial \lambda_{i}}$$

Some other variants include the slip-link model [86]:(67)$$W^{\mathrm{SL}}=\frac{1}{2}G_{e}\overset{3}{\underset{i=1}{\sum }}\left(\frac{\left(1+\eta \right)\left(1-\alpha^{2}\right)\lambda_{i}^{2}}{\left(1+\eta \lambda_{i}^{2}\right)\left(1-\alpha^{2}\sum_{i=1}^{3}\lambda_{i}^{2}\right)}+ln\left(1+\eta \overset{3}{\underset{i=1}{\sum }}\lambda_{i}^{2}\right)\right)$$
(68)$$\sigma_{11}^{\mathrm{SL}}{=\mu }_{c}\left({\lambda -\lambda }^{-2}\right)+\frac{{2\mu }_{e}}{\beta }\left(\lambda^{\frac{\beta }{2}-1}{-\lambda }^{-\beta -1}\right)$$
and also the tube [43] model. The tube model potential consists of two parts that characterize chain cross-linkings $W^{T,c}$ and chain entanglements $W^{T,e}$ such that
(69)$$W^{T}{=W}^{T,c}{+W}^{T,e}=\overset{3}{\underset{i=1}{\sum }}\frac{\mu_{c}}{2}\left({\bar{\lambda }}_{i}^{2}-1\right)+\frac{{2\mu }_{e}}{\beta^{2}}\left({\bar{\lambda }}_{i}^{-\beta }-1\right)$$
(70)$$\sigma_{11}^{T}{=\mu }_{c}\left({\lambda -\lambda }^{-2}\right)+\frac{{2\mu }_{e}}{\beta }\left(\lambda^{\frac{\beta }{2}-1}{-\lambda }^{-\beta -1}\right)$$

The most recent Gaussian-type model is a nonaffine tube model [87] characterized as
(71)$$W^{\mathrm{NT}}{=W}^{\mathrm{ph}}{+W}^{\mathrm{ent}}{=G}_{c}\overset{3}{\underset{i=1}{\sum }}\frac{\lambda_{i}^{2}}{2}{+G}_{e}\overset{3}{\underset{i=1}{\sum }}\left(\lambda_{i}+\frac{1}{\lambda_{i}}\right)$$
(72)$$\sigma_{11}^{\mathrm{NT}}{=\mu }_{c}\left({\lambda -\lambda }^{-2}\right)+\frac{{2\mu }_{e}}{\beta }\left(\lambda^{\frac{\beta }{2}-1}{-\lambda }^{-\beta -1}\right)$$

Non-Gaussian chain network models apply the mechanical property of a non-Gaussian single chain to the molecular chains of the rubber network. The first three-chain model assumed a distribution of molecular chains along principal directions and was efficient only for small stretches. This limitation was overcome by Arruda and Boyce by introducing limited chain extensibility. Furthermore, a more general distribution of molecular chains was allowed in further theories that also introduced chain entanglement and cross-linking.

The three-chain model is defined in the following manner [88]:(73)$$W^{3C}=\frac{\mu N}{3}\overset{3}{\underset{i=1}{\sum }}\left({\bar{\gamma }}_{i}{\bar{\lambda }}_{r,i}+\ln\left(\frac{{\bar{\gamma }}_{i}}{\sin h{\bar{\gamma }}_{i}}\right)\right)$$
(74)$$\sigma_{11}^{3C}=\frac{\mu }{3\lambda }\left(\lambda^{2}\left[\frac{{3N-\lambda }^{2}}{{N-\lambda }^{2}}\right]-\frac{1}{\lambda }\left[\frac{{3N-\lambda }^{-1}}{{N-\lambda }^{-1}}\right]\right)$$

The well-known and frequently used Arruda–Boyce model is presented next [89]:(75)$$W^{\mathrm{AB}}{=C}_{1}\left(\begin{array}{c} \left(\frac{1}{2}I_{1}-3\right)+\frac{1}{20{\left(\lambda_{A}\right)}^{2}}+\frac{1}{1050{\left(\lambda_{A}\right)}^{4}}\left(I_{1}^{3}-27\right)+\\ \frac{19}{7000{\left(\lambda_{A}\right)}^{6}}\left(I_{1}^{4}-81\right)+\frac{519}{673750{\left(\lambda_{A}\right)}^{8}}\left(I_{1}^{5}-243\right) \end{array}\right)\;{=C}_{1}\overset{5}{\underset{i=1}{\sum }}\alpha_{i}{\hat{\beta }}^{i-1}\left(I_{1}^{i}-3^{i}\right)$$
(76)$$\sigma_{11}^{\mathrm{AB}}{=2C}_{1}\left(\lambda -\frac{1}{\lambda^{2}}\right)\left(\begin{array}{c} \frac{1}{2}+\frac{2}{20{\left(\lambda_{A}\right)}^{2}}\left(I_{1}\;-9\right)+\frac{3}{1050{\left(\lambda_{A}\right)}^{4}}\left(I_{1}^{2}-27\right)+\\ \frac{4\cdot 19}{7000{\left(\lambda_{A}\right)}^{6}}\left(I_{1}^{3}-81\right)+\frac{5\cdot 519}{673750{\left(\lambda_{A}\right)}^{8}}\left(I_{1}^{4}\;-243\right) \end{array}\right)$$

The eight-chain model is as follows [90]:(77)$$W^{8C}=\frac{\mu N}{3}\left(\bar{\gamma }{\bar{\lambda }}_{r}+\ln\left(\frac{\bar{\gamma }}{\sin h{\bar{\gamma }}_{i}}\right)\right)$$
(78)$$\sigma_{11}^{8C}=\frac{\mu }{3}\left(\left[\frac{{3N-\lambda }_{\mathrm{cu}}^{2}}{{N-\lambda }_{\mathrm{cu}}^{2}}\right]\right)\left({\lambda -\lambda }^{-2}\right),\;\lambda_{\mathrm{cu}}=\sqrt{\frac{1}{3}\left(\lambda +\frac{2}{\lambda }\right)}$$

A systematical consideration of the affine, three-chain, eight-chain, and micro-sphere models is provided in [91].

The Flory–Erman approach, which is an extension of the nonaffine-tube model and consists of the phantom energy $W^{\mathrm{FE},\mathrm{pe}}$ function and micromechanics of chain molecules $W^{\mathrm{FE},\mathrm{ce}}$, is provided in the following way [92]:(79)$$W^{\mathrm{FE}}{=W}^{\mathrm{FE},\mathrm{pe}}{+W}^{\mathrm{FE},\mathrm{ce}}=\overset{3}{\underset{i=1}{\sum }}\frac{\mu }{2}\left(\left[1-\frac{1}{\phi }\right]\left[{\bar{\lambda }}_{i}^{2}-1\right]+\left[B_{i}{+D}_{i}-\ln\left(B_{i}+1\right)-\ln\left(D_{i}+1\right)\right]\right)\;\xi =\left[1-\frac{1}{\phi }\right]{n,\;\mu =\mathrm{nK}\theta ,\;B}_{i}{=\kappa }^{2}\left({\bar{\lambda }}_{i}^{2}-1\right){\left({\bar{\lambda }}_{i}^{2}+\kappa \right)}^{-2},\;D_{i}={\bar{\lambda }}_{i}^{2}\kappa^{-1}B_{i}$$
(80)$$\sigma_{11}^{\mathrm{FE}}=\mu \left(1-\frac{2}{\phi }\right)\left({\lambda -\lambda }^{-2}\right)+\frac{\partial U_{\mathrm{FE},\mathrm{ce}}^{m}}{\partial \lambda_{i}}$$

The Gaussian nature of the phantom part of this model causes a deviation at high strains. This is overcome in its modified version proposed by Boyce and Arruda. It replaces the neo-Hookean part with an eight-chain model such that
(81)$$W^{\mathrm{FE},m}{=W}^{8C}{+W}^{\mathrm{FE},\mathrm{ce}}=\mu N\left(\bar{\gamma }{\bar{\lambda }}_{r}+\ln\left(\frac{\bar{\gamma }}{\sin h{\bar{\gamma }}_{i}}\right)\right)+\overset{3}{\underset{i=1}{\sum }}\frac{\mu }{2}\left(\left[B_{i}{+D}_{i}-\ln\left(B_{i}+1\right)-\ln\left(D_{i}+1\right)\right]\right)$$
(82)$$\sigma_{11}^{\mathrm{FE},m}{=\sigma }_{11}^{8C}+\frac{\partial W^{\mathrm{FE},\mathrm{ce}}}{\partial \lambda_{i}}$$

In its simplified version, the micro-sphere model [93] is given as
(83)$$W^{\mathrm{MS}}=\mu \left(\lambda_{r}L^{-1}\left(\lambda_{r}\right)+\ln\left(\frac{\gamma }{\sin h\left(\gamma \right)}\right)\right)$$
(84)$$\sigma_{11}^{\mathrm{MS}}{=\mu }_{c}\left({\lambda -\lambda }^{-2}\right)+\frac{{2\mu }_{e}}{\beta }\left(\lambda^{\frac{\beta }{2}-1}{-\lambda }^{-\beta -1}\right)$$

One of the modern formulations of a hyper-elastic material is the extended tube model [94], which includes five parameters. It focuses on the molecular–statistical approach for polymer networks and it is based on the following formulae: (85)$$W^{\mathrm{ET}}=\frac{\mu_{c}}{2}\left(\frac{\left({1-\delta }^{2}\right)\left({\bar{I}}_{1}-3\right)}{{1-\delta }^{2}\left({\bar{I}}_{1}-3\right)}+\ln\left({1-\delta }^{2}\left({\bar{I}}_{1}-3\right)\right)\right)+\overset{3}{\underset{i=1}{\sum }}\frac{{2\mu }_{e}}{\beta^{2}}\left({\bar{\lambda }}_{i}^{-\beta }-1\right)$$
(86)$$\sigma_{11}^{\mathrm{ET}}{=\mu }_{c}\left({\lambda -\lambda }^{-2}\right)\left(\frac{{1-\delta }^{2}}{{\left({1-\delta }^{2}\left(I_{1}-3\right)\right)}^{2}}-\frac{\delta^{2}}{{1-\delta }^{2}\left(I_{1}-3\right)}\right)+\frac{{2\mu }_{e}}{\beta }\left(\lambda^{\frac{\beta }{2}-1}{-\lambda }^{-\beta -1}\right)$$

Network averaging tube [95]:(87)$$W^{\mathrm{NA}}{=\mu }_{c}\kappa n\;\ln\left(\frac{\sin\left(\frac{\Pi }{\sqrt{n}}\right){\left(\frac{I_{1}}{3}\right)}^{q/2}}{\sin\left(\frac{\Pi }{\sqrt{n}}{\left(\frac{I_{1}}{3}\right)}^{q/2}\right)}\right)$$
(88)$$\sigma_{11}^{\mathrm{NA}}=\frac{\mu }{3}\left(\left[\frac{{3N-\lambda }_{\mathrm{cu}}^{2}}{{N-\lambda }_{\mathrm{cu}}^{2}}\right]\right)\left({\lambda -\lambda }^{-2}\right)$$SpT [96]:(89)$$W^{\mathrm{SpT}}=G_{c}\mathrm{Nln}\left(\frac{3N+\frac{1}{2}I_{1}}{3N-I_{1}}\right)+G_{e}\underset{i}{\sum }\lambda_{i}$$
(90)$$\sigma_{11}^{\mathrm{SpT}}=\frac{\mu }{3}\left(\left[\frac{{3N-\lambda }_{\mathrm{cu}}^{2}}{{N-\lambda }_{\mathrm{cu}}^{2}}\right]\right)\left({\lambda -\lambda }^{-2}\right)$$Wu-Giessen or full-network [97]:(91)$$W^{\mathrm{FN}}{=W}^{3C}\left(1-\rho \right){+\rho W}^{8C}$$
(92)$$\sigma_{11}^{\mathrm{FN}}{=\sigma }_{11}^{3C}\left(1-\rho \right){+\rho \sigma }_{11}^{8C}$$Lim [98]:(93)$$W^{L}=W^{G}\left(1-f\right)+fW^{8C}$$
(94)$$\sigma_{11}^{S}=\left(\frac{1}{2}\overset{N}{\underset{p=1}{\sum }}A_{p}{\left(\frac{I_{1}}{3}\right)}^{\alpha_{p}}+\frac{1}{2}\overset{N}{\underset{p=1}{\sum }}B_{p}{\left(\frac{I_{1}}{3}\right)}^{\beta_{p}}\right)\left(\lambda -\frac{1}{\lambda^{2}}\right)$$Bechir and Chavalier [99]:(95)$$W^{\mathrm{WG}}=\frac{3}{2}\overset{N}{\underset{p=1}{\sum }}\frac{A_{p}}{{1+\alpha }_{p}}{\left(\frac{I_{1}}{3}\right)}^{{1+\alpha }_{p}}+\frac{3}{2}\overset{N}{\underset{p=1}{\sum }}\frac{B_{p}}{{1+\beta }_{p}}{\left(\frac{I_{1}}{3}\right)}^{{1+\beta }_{p}}$$
(96)$$\sigma_{11}^{S}=\left(\frac{1}{2}\overset{N}{\underset{p=1}{\sum }}A_{p}{\left(\frac{I_{1}}{3}\right)}^{\alpha_{p}}+\frac{1}{2}\overset{N}{\underset{p=1}{\sum }}B_{p}{\left(\frac{I_{1}}{3}\right)}^{\beta_{p}}\right)\left(\lambda -\frac{1}{\lambda^{2}}\right)$$

Only recently have some efforts been made to define the hyper-elastic behavior of multi-phase materials. This was because of the high complexity of numerical simulations and the unavailability of analytical solutions. These problems were overcome by a rapid increase in computational resources, allowing for an iterative solution. Some examples of contemporary studies include [100,101,102,103,104,105,106,107,108], as well as [109] for the visco-elastic regime. Demand for advanced hyper-elastic models is stimulated by the biotechnological industry, where the mechanics of biological tissue [110] and its interaction with artificial appliances were studied [21]. They are also required in aerial, textile, and automotive industries, and are used, for example, in tires or various shock absorbers. It needs to be mentioned that Poisson ratio close to the value of 0.5 for rubber-like materials may break the FEM solution incremental processes (in both deterministic and probabilistic context) due to the negative diagonal components within the stiffness matrix or a lack of numerical convergence. 

Unlike in linear small strain elasticity, all the models provide their own set of assumptions that were prescribed during the derivation process and are not necessarily easily interchangeable with others. This introduces challenges in the description of their stochastic nature because the material parameters are not a good choice for overall stochastic unknowns. Such a choice would bind the analysis to a specific model, which is highly undesirable. Instead, some more generic parameters or variables should be preferred, for example, a defects volume fraction, which is unique for all models; random (effective) material coefficients should instead be derived from other random sources.

Stochastic or probabilistic studies are quite well documented in the reversible elastic regime of composites [111,112]. This is not true for the inelastic regime. One of the main reasons is the abovementioned lack of generality of various laws and the complexity of their numerical solution already given in the deterministic case. The stochastic hyper-elasticity or visco-elasticity of composites is considered only in a limited number of studies; some examples include [113,114,115,116]. Much attention was focused on a certain hyper-elasticity potential, for example, the Ogden [117,118,119], neo-Hookean [66,120], or van der Waals models [105]; some of these also compare the results of several potentials, as in [121], or propose a specific solution technique [122]. However, there is a lack of studies concerning the composites that include interface defects in the nonlinear regime. This is especially true for a joint deterministic and stochastic analysis coupled with the verification of multiple hyper-elastic potentials. Unlike the majority of the stochastic considerations, these analyses are additionally based on the set of laboratory tests performed especially for the computational part, for which the numerical response of the matrix is fitted with the use of the least squares method. The proposed approach specifically tackles the problem of the lack of generality of hyper-elastic laws via the introduction of a probabilistic homogenization algorithm for probabilistic homogenization. It enables the computation of random effective material parameters for an arbitrary linear hyper-elastic potential with a specified source of the input uncertainty.

Please note that the engineering stress $\sigma_{i}$ for the above models can be obtained for all the principal directions i using the following formulae:(97)$$\sigma_{i}=\frac{\partial W}{\partial I_{1}}\frac{\partial I_{1}}{\partial \lambda_{i}}-\frac{\partial W}{\partial I_{2}}\frac{\partial I_{2}}{\partial \lambda_{i}}-\frac{1}{\lambda_{i}}p$$
and thus, for uniaxial tension, $\sigma_{1}^{\mathrm{UT}}=2\left(\frac{\partial W}{\partial I_{1}}+\frac{1}{\lambda }\frac{\partial W}{\partial I_{2}}\right)\left(\lambda -\frac{1}{\lambda^{2}}\right)$ and $\sigma_{2}^{\mathrm{UT}}{=\sigma }_{3}^{\mathrm{UT}}=0$; for pure shear, $\sigma_{1}^{\mathrm{PS}}=2\left(\frac{\partial W}{\partial I_{1}}+\frac{\partial W}{\partial I_{2}}\right)\left(\lambda -\frac{1}{\lambda^{3}}\right)$, $\sigma_{2}^{\mathrm{PS}}=2\left(\frac{\partial W}{\partial I_{1}}{+\lambda }^{2}\frac{\partial W}{\partial I_{2}}\right)\left(\lambda -\frac{1}{\lambda^{2}}\right)$, and $\sigma_{3}^{\mathrm{PS}}=0$; and for equi-biaxial tension, $\sigma_{1}^{\mathrm{BT}}=2\left(\frac{\partial W}{\partial I_{1}}+\frac{1}{\lambda }\frac{\partial W}{\partial I_{2}}\right)\left(\lambda -\frac{1}{\lambda^{5}}\right)$ and $\sigma_{3}^{\mathrm{BT}}=0$.

Artificial neural networks emerged in constitutive model determination 30 years ago [123]. Instead of first selecting its closed form and then the tuning parameters, they propose a family of artificial neural networks and then learn its weight and parameters; they may be physical, phenomenological, or mixed. Until recently, classical neural networks remained a black box in terms of the morphology of parameters and partially or totally disregarded the kinematic, thermodynamic, and physical constraints. They also bypassed constitutive modeling altogether. This raised well-posed concerns for their formulation. These were covered recently via the direct inclusion of the physical and mechanical constraints into the neural network [124,125,126].

A very interesting innovation in this regard constitutes artificial neural networks. They take an a priori closed set of constitutive models whose distribution in the final formula is defined during the learning process [127,128]; such a formulation overcomes the above concerns. Such a trained network includes a sum-of-weighted closed-form models for each loading scheme. Thus, it is not strictly a new model in itself but is rather a novel mixture of already created ones; its successful utilization requires prior knowledge of classical models and cannot work without them. 

## 3. Probabilistic Methods

The structural behavior of materials and structures in civil and mechanical engineering design is defined by models with specific assumptions. Traditionally they have a certain set of parameters to fit them to the response of the material in well-known tests. These parameters are always exposed to some scatter, which comes from various sources of randomness, such as the morphology of the material, tolerances in manufacturing and measurements, accuracy of conversions, or other unknown origins that are impossible to quantify during measurement.

Probabilistic analysis is an approach that tackles this problem directly by augmenting deterministic mathematical models with random parameters or variables that represent sources of uncertainty. It allows for a much more precise estimation of the behavior of the analyzed system. In addition to the mean, characteristic, or design values obtained in most engineering calculations, it also outputs the expected value, coefficient of variation, and other specific information about the uncertainty of the results. It further allows us to directly compute and optimize the structural safety margin in certain load conditions. This margin may be represented, for example, as a reliability index or probability of survivability. Probabilistic analysis may be applied in almost all (and not only) structural analyses at various levels of design, i.e., the level of material, structural element, or even the level of the entire structure; recent examples include heat transfer [129,130], fatigue [131,132], stiffness [133], failure [134,135,136], or system response under uncertainty [137].

The probabilistic approach encompasses all the methods based on probability calculus, leading to the calculation of the material or structural response with input uncertainties. This response is commonly represented as random moments or characteristics, such as the expected value, coefficient of variation, skewness, or kurtosis [138]. The most precise representation of statistical distribution is the probability density function (PDF), but it is also the most demanding in terms of computational resources. This is why in most cases, probabilistic characteristics are preferable, as in the study of hyper-elastic materials. The most widespread methods that allow for probabilistic design are direct derivation methods, simulation methods, spectral methods, and perturbation methods.

Direct derivation methods [14,138] use integral calculus to derive random characteristics of the response with a known PDF of the input random variables. In its classical form, this approach uses a direct relation between the structural response and the random parameter; it is commonly called an analytical method (AM). In many cases, such a relation is not known, cannot be derived analytically, or a symbolic solution for the known relation simply does not exist. An approximation of this relation is obtained with various numerical techniques that are commonly called a response function or a response surface. Such an approach is called a semi-analytical method (SAM) [138,139] and is applied in the study of hyper-elastic materials; alternatively, some approximate integration may be employed.

Computer simulation methods [140,141,142,143,144,145] substitute integration with a finite number of deterministic realizations, which are then subjected to statistical estimation. Realizations are made with sets of parameters obtained from random or pseudo-random generation according to their predefined PDFs; these are all referred to as Monte Carlo simulation (MCS) methods [140,146,147]. They have some advantages, i.e., ease of implementation and avoidance of integration, but they all also have important disadvantages. The major drawback of this approach is the vast number of realizations required to reach a satisfactory convergence result, which is guaranteed only with their infinite number [140,148]. This problem is partially covered in more modern approaches, such as importance sampling [149], stratified sampling [150], or Latin hypercube sampling [151] techniques; all of these approaches attempt to lower the required sampling number. A second disadvantage is poor scaling for an increasing number of unknowns. The last and deciding disadvantage is that the inherent discrete character of the MCS disables the retrieval of continuous probabilistic characteristics or measures of reliability.

The spectral method [130,152,153,154] describes the Gaussian random field with the use of the Karhunen–Loève expansion. The structural response is emulated by expansion into polynomial chaos [155], but the biggest problem of this methodology is the number of terms in the expansion series necessary to achieve satisfactory accuracy [156]. Because of this, the result may be inadequate and quite far from the exact solution, especially when a second-order polynomial is used (which is commonly done). For this expansion to obtain satisfactory accuracy, a high number of elements may oftentimes be required. An additional problem is connected to the Karhunen–Loève expansion itself, which does not have a solution for certain problems [157,158], with an especially important example being a reliable calculation of higher-order probabilistic characteristics [159].

Perturbation methods [138,159,160] describe the structural response as spread around its mean value with a given small perturbation. They expand the response function into a Taylor series around the expected values of random variables. The order of this method depends on the number of terms in the applied expansion, namely, one term in the first order, two terms in the second order, etc. One of its issues is the convergence of the Taylor series, which is not always guaranteed. This problem is even more complex when the response function is not known and must be approximated, which is the case for the study of hyper-elastic materials. The response function is selected from a set of polynomial functions, where the theoretical convergence of specific Taylor series can be mathematically proven. The major advantages of this method are swift execution, relative ease of implementation, and continuous character of the results. It also substitutes integrals with derivatives and overcomes the problem of the lack of an analytical solution. The biggest drawback is the probabilistic convergence for certain types of functions and the requirement of a higher order expansion to reach high accuracy for input random variables with a high coefficient of variation. A relation between the result and the input random variable is usually unknown before the probabilistic analysis. Its analytical derivation is available only for relatively simple, well-known problems. In other cases, this relation is sought with the use of numerical methods, such as the finite element method, boundary element method, and finite difference method; joining their output with probabilistic analysis results in the stochastic finite element method (SFEM) [153,161,162,163,164], stochastic boundary element method (SBEM) [165,166], and stochastic finite difference method (SFDM) [138,167]. Such a fusion creates a powerful tool for probabilistic design and stochastic computations but it does not provide a new probabilistic method itself. This is because of the convenience of solving the homogenization problem and the availability of software for FEM computations; readers looking for a comprehensive introduction to the finite element method may refer to [168].

## 4. Interphase and Interface Defects

An interphase is an additional phase of the composite formed during its manufacturing process or exploitation. It is introduced (artificially in modeling or during manufacturing) in between two phases of a composite, commonly between the matrix and the filler (reinforcement). Its mechanical [169], thermal, electrical, thermo-mechanical [170], and physical characteristics differ from the ones of the two surrounding phases. Its volume is much lower than the other phases, yet it highly influences the behavior of an entire composite. This is because the interphase effectively encapsulates the filler and prevents a direct interaction in between the composite constituents. As documented, an interphase significantly affects the effective material properties of multiple isotropic, cubic, and anisotropic composites in a deterministic [171,172,173,174,175] and also stochastic context [8,11,176,177]. It either decreases them in the existence of defects [178] or increases when the two phases are chemically bound [179,180,181,182]. Its influence is so high that considerable research attention has been put toward its tuning and tweaking to improve key properties of composites or adjust their performance for special purposes [183,184].

An interphase is extremely difficult to localize and further analyze in laboratory tests. This is because its thickness is very small (in the order of micrometers) and its characteristics differ for each particle or fiber in the same sample of the composite. Despite the existence of multiple surface agents applied in the phases, the interphase around each filler particle or fiber has unique thermo-electro-mechanical conditions in which it forms; this effectively causes its geometry, thickness, and properties to be random (see [185]).

Interface defects encompass all the inclusions, voids, discontinuities, and inaccuracies that exist in the transition of the two phases in a composite. They reflect frequent manufacturing imperfections, for instance, following significant residual thermal stresses, and can be treated during a numerical simulation as geometrical imperfections in composite materials [186]. Their occurrence greatly affects the response and properties of the composite despite its extremely small volume, which is a fraction of the volume of the interphase. Interface defects were shown to be crucial for many properties of a composite, including its durability [187], reliability [188], thermal conductivity [189], and even failure [180,190,191,192]. The defects and inclusions also cause a high microscopic stress concentration [193]. Some studies were devoted to the defects only [194].

An interphase with interface defects forms an imperfect interface. Its usage in the realistic prediction of composites’ behavior dates back many years [195,196,197,198,199]. The analysis of composites with imperfect interfaces is performed primarily with the use of three techniques: (1) insertion of the interphase in between the main composite constituents [200,201,202,203,204], (2) usage of special contact finite elements [205,206], and (3) an application of a system of the springs [207] that may also be supplemented with dumpers. Very interesting strategies belonging to the first group are based upon a geometrical idealization of such defects, e.g., with the use of semi-circular or semi-spherical shapes, which follows the well-known cavitation phenomenon for a variety of matrices [190,208].

## 5. Multiscale Models and Numerical Simulation

Multiscale analysis represents some trends in current numerical modeling in which the given heterogeneous system is described simultaneously by multiple models at different scales of resolution. Models at each scale may originate from physical laws of different natures, for example, continuum mechanics at the macroscale [209] and molecular dynamics at the atomistic scale. They are required because certain phenomena visible at one scale cannot be described accurately without supplementary information from a different scale from which they originate. Some examples include (1) brittle failure of the reinforced concrete beam, which is caused by micro-cracks in concrete microstructure in between the cement and grains; (2) plastic elongation of steel caused by a slip between its grains; or (3) macroscopic properties of composites affected by microstructural defects. Alternatively, they are used because purely macroscale models are not accurate enough, while lower-scale models offer too much information or require too high computational power to be executed. The multiscale approach aims at achieving a compromise between accuracy and efficiency and frequently provides solutions to problems that are otherwise unsolvable in a reasonable period. The multiscale analysis comprises three major components: multiscale models, multiscale analysis tools, and multiscale algorithms. Multiscale algorithms involve using multiscale analysis tools to bridge the scales and connect different multiscale models at different levels.

The most used scales depend mainly upon the dimensions and are the following: macroscale, mesoscale (level of microstructure), atomistic scale, and electronic scale. Interestingly, each of these scales falls into a different discipline, for example, the macroscale falls into civil and mechanical engineering, the mesoscale falls into materials science, and the electronic scale falls into physics. Thus, multiscale design frequently requires bridging different disciplines to be solved. The connection between the models is either ensured analytically or numerically [210,211,212] and bridging between different scales is either done sequentially or concurrently. In the sequential approach, certain characteristics used in macroscopic models are precomputed. Usually, only a limited number of the parameters (or variables) is passed to a different scale, but in some sparse representations, even as many as six variables can be passed effectively [213,214]. In the concurrent alternative, the macroscale model variables are computed on the fly during simulation. Its major advantage is that a much smaller domain of macroscopic variables must be computed on a different scale. On the other hand, the complexity of numerical algorithm solving macroscale increases. Concurrent methods are not especially well suited to problems in which parameters are passed to the FEM-based code, where each element would potentially require a separate set of parameters.

Traditional structural analysis preferred in civil engineering does not involve a multiscale approach because it tends to limit calculations to a linear range of material deformation, where simple empirical constitutive laws are sufficient. The additional margin of a nonlinear structural response is frequently used as a safety margin. A good example is the design of structures made from constructional steel, for which the engineering codes, such as Eurocodes, prefer the usage of linear elastic models. Calculations become much more involved (even for steel) when localized phenomena, such as strain localization [215] or crystal plasticity [216], must be taken into consideration. For these phenomena, a macroscale model must be supplemented or interchanged with a macroscopic structure. Macroscopic stress in the FEM crash test is calculated with supplementary information from the mesoscale (scale of crystals and phases), as well as lower scales, where defects originate. Unlike in statics, in structural dynamics, a scale is not only defined for space, but also for time. The smaller the scale, the lower the dimensions and the smaller the timespan. In multiscale design, information may be passed from a higher scale into a lower scale (top-down method) or from a lower to a higher scale (bottom-up method).

The process of crossing a certain scale is called scale-bridging; an example of top-down bridging is passing boundary conditions for each element of the FEM analysis into a mesoscale model, while a bottom-up bridging example involves intra-grain bonding conditions coming from the atomistic scale. A multiscale approach is already used, albeit indirectly, at the structural design level for reinforced concrete. Verification of the ultimate limit state for this composite already requires some supplementary information apart from the scale of the structural element. Specifically, precise information on steel rebar positioning is required even when all the macroscopic properties of concrete and steel are already known. These properties already depend on information coming from four lower scales, i.e., C-S-H, cement paste, mortar, and concrete mesoscale scales [217].

There exist multiple methods and algorithms that allow for solving multiscale problems. They focus on algebraic, numerical, or hybrid solutions and are usually aimed at certain problem domains. Their comprehensive review is available in [218]. Some examples include the multi-grid methods aimed at solving a large system of algebraic equations [219] with some alteration in the equation-free method [220] and a heterogeneous multiscale method where a preconceived macroscale model with missing components is assembled and missing data is found with the use of microscale models or matched asymptotics approach [221,222,223]. Some other methods include averaging and tolerance-averaging methods [224,225], hydrodynamic limit methods [226], the Mori–Zwanzig formalism [227], renormalization group methods [228], variational methods [229], and homogenization methods [230,231,232,233,234,235,236]. The multiscale approach is very frequently used for the determination of the macroscopic (or effective) properties of composites, also in the stochastic context [237,238]. Some introduction to their theory is available, for example, in [239,240]. The homogenization method is often chosen because of its relatively easy application in FEM systems.

## 6. Homogenization Method and Effective Medium Response

The homogenization method is frequently referred to as a process of the replacement of an equation with a highly oscillatory coefficient with one that has a homogenous coefficient. Initially used in studying partial differential equations (PDEs) [230,241], this concept was found to be efficient at solving a problem of the non-homogenous microstructure of continua, such as composites. This is because the macroscale boundary conditions, such as forcings, loadings, or supports, are much bigger than the length scale of the microstructure. A classical boundary value problem is given in the following way:(98)$$\nabla \cdot \left(C\left(\frac{\vec{x}}{\tau }\right)\nabla u_{\tau }\right)=f$$
with τ being a very small parameter and $C(\vec{g})$ being a periodic coefficient $C=(\vec{g}+{\vec{e}}_{j})=C(\vec{g}),\;i=1,\;\ldots ,\;n$. It may be modified to the following form:
(99)$$\nabla \cdot \left(C^{*}\nabla u\right)=f$$
where $C^{*}$ is the effective constitutive tensor with constant coefficients representing a homogenized material. This property could be computed as
(100)$$C_{\mathrm{ij}}^{*}=\int_{{\left(0,1\right)}^{n}}C\left(\vec{g}\right)\left(\nabla w_{j}\left(\vec{g}\right)+{\vec{e}}_{j}\right)\cdot {\vec{e}}_{i}{\mathrm{dy}}_{1}{\ldots \mathrm{dy}}_{n},\;\;i,j=1,\ldots ,n$$
where periodic $w_{j}$ satisfies $\nabla_{y}(C(\vec{g})\nabla w_{j})=-\nabla_{y}\cdot (C(\vec{g}){\vec{e}}_{j})$. One equation may be replaced by another if τ is small enough to satisfy $u_{\tau }\approx u$ and when $u_{\tau }\to u$ at τ→0. Regarding the continuum concept, an analog of the differential element is the representative volume element (RVE) in 3D problems or the representative surface element (RSE) in 2D problems. This should be selected in a way to contain all the relevant statistical information about an inhomogeneous medium. With such an assumption, averaging over this element results in an effective property of the medium defined as $C^{*}$ above. A key problem in such a formulation is the assumption that such an RVE is solvable and contains as much information about the microstructure as possible. This also holds for stochastic calculations where, in addition to the RVE selection, uncertain parameters must also be selected in a way to catch the best representation of the most important randomness sources and remain simple enough to be solved.

In its early approaches, homogenization was used in a purely analytical [233,242] way, and thus, the microstructure was very simple and the range of applications was limited. This obstacle was overcome with the incorporation of the FEM, where the RVE was modeled and solved. The rapid evolution of the academic and commercial FEM software that started at the end of the twentieth century supported researchers in the discretization and visualization of the RVE. A simultaneous revolution in the computational power of personal computers made solutions to more complex problems accessible. All of this allowed homogenization to become one of the most widespread methods used to solve multiscale problems in materials science, especially those connected to the meso- and microscale of inhomogeneous materials (see, e.g., [243,244,245]). The topic of a correct RVE is so important that some studies treat it as a research problem connected with its generation [246], size or scale effect [247,248,249,250,251], and validity of applied random microstructure [252]. Some other works reviewed existing stochastic boundary conditions and introduce new ones [253] or propose a formulation of finite elements for the hyper-elastic case [254]. The FEM is commonly included in approaches that are aimed at decreasing the problem size or computation time, which is done at the expense of accuracy. These propose the usage of artificial neural networks and machine learning approaches [255,256,257], divide the heterogeneous medium into several subdomains [117], use a manifold-learning method to reduce the dimensions of microscopic strain fields [258], utilize orthogonal decomposition R3M [259], or aim to apply a reduced database model [260], to name a few. The FEM is, of course, not the only possibility. There also exist some alternative methods, such as mesh-free formulations [261], a gradient approach [262], fast Fourier transform (FFT)-based methods [263], a transformation field [264], or the discrete element method [265], which are perfect for densely packed solids. Sometimes, a nonlinear response of the material is studied, for which some approaches to homogenization exist, such as the ones in [266,267,268,269]. In many cases, they include specific a priori assumptions related to the stress or strain fields (see, e.g., [270,271,272]). Probabilistic homogenization adds yet one more level of abstraction to homogenization. It introduces uncertain parameters and variables in the lower scale of homogenization, usually in the material microstructure; some examples were presented and discussed in [273,274,275,276,277,278,279]. Similarly to probabilistic analysis in macroscopic calculations, it quantifies the influence of input uncertainty on the response of the medium. The difference is the source of uncertainty, which cannot be included at a macroscopic level, yet cause randomness in engineering structures. Common problems for which probabilistic homogenization is applied include uncertain phase properties, interface defects, geometric uncertainty, or inclusions. Uncertainty may be included in one phase or in in the various characteristics of the RVE, for example, the reinforcement positioning. They all result in an uncertain stiffness tensor or parameters leading to the material constitutive relation.

Probabilistic homogenization of composites is especially interesting when it is coupled with the problem of interface defects. An analytical solution to this problem can only be obtained for elastic composites and a simple RVE, which was proposed in [280]. A more in-depth analysis of the stiffness tensor, even in an elastic regime, requires the usage of numerical solvers. The author proposed such an approach [204], verified it with an analytical solution, and studied a fully anisotropic response of the homogenized composite [11]. The results demonstrated that particle clustering and uneven particle distribution affect the anisotropy of the composite and have a high influence on the components of its stiffness tensor. In his other work [8], he verified a numerical solution of a composite with an uncertain reinforcing particle radius with an analytical solution and studied an influence of an uncertain aspect ratio on the effective stiffness tensor of a composite. One may extend this model toward a hyper-elastic regime of the composite, where an uncertain hyper-elastic response of a homogeneous medium can be taken into account [115], which may be further extended toward the stochastic hyper-elastic response of composites with hysteresis [99] and with stochastic interface defects [281,282].

As it is well-known, the effective response of a medium is a relation of the objective function with uncertain parameters or variables. In structural design, the objective function could be defined as a limit function. In homogenization, it usually is an effective property of a medium, such as the stiffness tensor, bulk modulus, effective stress, or strain energy. In the majority of homogenization problems (and also in most structural engineering problems), such a relation cannot be analytically determined. This is a reason why the objective function is commonly computed with the use of discrete numerical procedures. In SPT, this is frequently done using a response function method (RFM) [138] or response surface method (RSM) [283,284] when more variables are considered. An alternative to the direct differentiation method (DDM) is rarely selected because it requires at least an intervention into a source code of a discrete numerical solver or even the introduction of its solver. This is because the deterministic values used by these programs must be substituted with their stochastic counterparts.

The response function method and the response surface method both aim to approximate the real relations of the objective function with the use of a surrogate model (also called a meta-model [285,286]) with an uncertain variable. This is done based on a carefully selected set of discrete numerical (or laboratory) experiments performed for different values of input variables [287,288]. Their major advantage is the ease of application and disconnection of the metamodel fitting from the stochastic procedure, which allows for a simple analysis of the fitting errors. Surrogate models are commonly applied from a subset of polynomial functions and also their fractions or other rational functions [289,290]; a little less common is an application of the B-spline, logarithmic, exponential, or hyperbolic functions.

Searching for the response functions or surfaces is an optimization problem and can be solved using linear programming (LP), quadratic programming (QP), and nonlinear programming (NP) methods. A class of linear programming problems is one where the objective function and all of the constraints are linear functions of the decision variables. It always has either (a) one or more equivalent globally optimal solutions, (b) has an unbounded objective, or (c) no feasible solution. It is convex and has at most one feasible region with “flat faces” (i.e., no curves) on its outer surface. Its optimal solution (if available) lays at a “corner point” on this surface that is represented by constraints. A solver may work pointwise and the solution is fast. Common solvers include families of the simplex technique in its primal [291] or dual [292] version and the interior point technique [293], where the fitting procedure is completed with a linear or (sometimes) nonlinear version of the Least Squares Method.

As it is well known, a quadratic programming problem has an objective function that is a quadratic function of the design variables and constraints that are all linear functions of the variables. They have only one feasible region with “flat faces” on its surface (due to the linear constraints), but the optimal solution may be found anywhere within this region or on its surface. As it is well known, an objective function may be convex or non-convex. The convex functions have either positive definite or semi-definite Hessians, and non-convex have an indefinite Hessian and a saddle shape, which is usually out of the scope of QP solvers. Typical solvers include a simplex extension to the QP, active set and working set method variations [294,295], interior point [296,297,298], or Newton barrier methods [299,300,301,302]. A nonlinear programming problem similar to the hyper-elastic stochastic analysis of composites and their effective characteristics is no doubt one of the most difficult issues in optimization theory. An objective function is generally a nonlinear function of the decision variables and may have many locally optimal solutions. A global minimum is very difficult to be found [303], and common solvers include augmented Lagrange methods [304], sequential quadratic programming [305,306], and reduced gradient methods [307,308].

## 7. Concluding Remarks

As demonstrated in this review work, an application of probabilistic and stochastic methods in the analysis of hyper-elastic solids still attracts many researchers and has a large audience. Such an approach is especially very convenient for engineering practice with polymer-based composites, specifically elastomers. Experimental statistics included in new theoretical and computer models allow for further optimization of such materials. This may be crucial for the optimal design of structural dampers, whose reliability, durability, and structural health monitoring may bring huge qualitative and quantitative savings. Stochastic multiscale models, especially including the atomistic scale of the solid, are indeed still very scarce, mainly since this scale uncertainty requires relatively expensive Monte Carlo simulations to obtain reliable stochastic analyses. Time and computer power consumption in this case is still slowly decreasing, but this is the main reason to continuously look for some concurrent techniques. This is also why various homogenization methods remain very attractive in materials engineering, especially those accounting for material and geometrical imperfections of a random nature, and most probably will remain so in the future.

## Figures and Tables

**Figure 1 materials-15-08878-f001:**
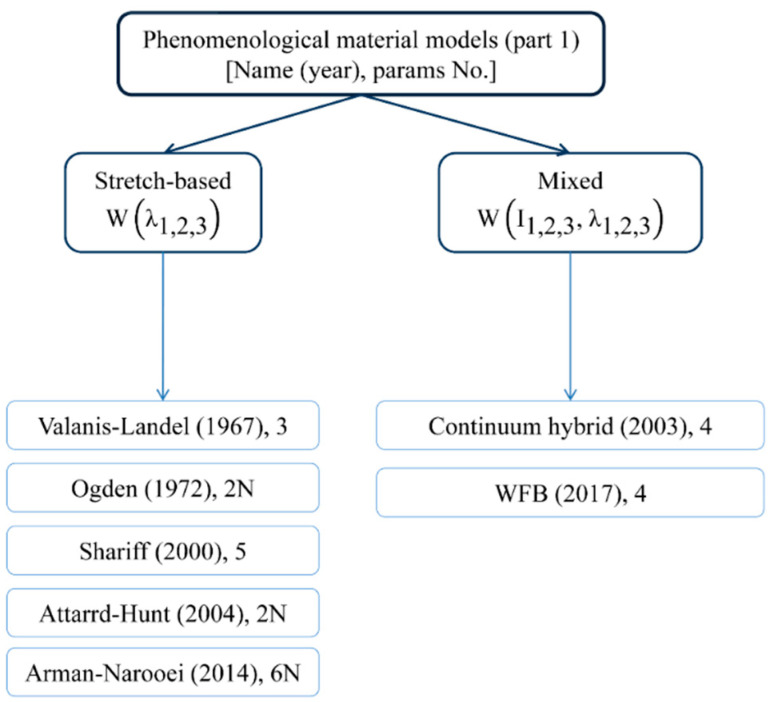
Different phenomenological models of hyper-elastic materials (part 1).

**Figure 2 materials-15-08878-f002:**
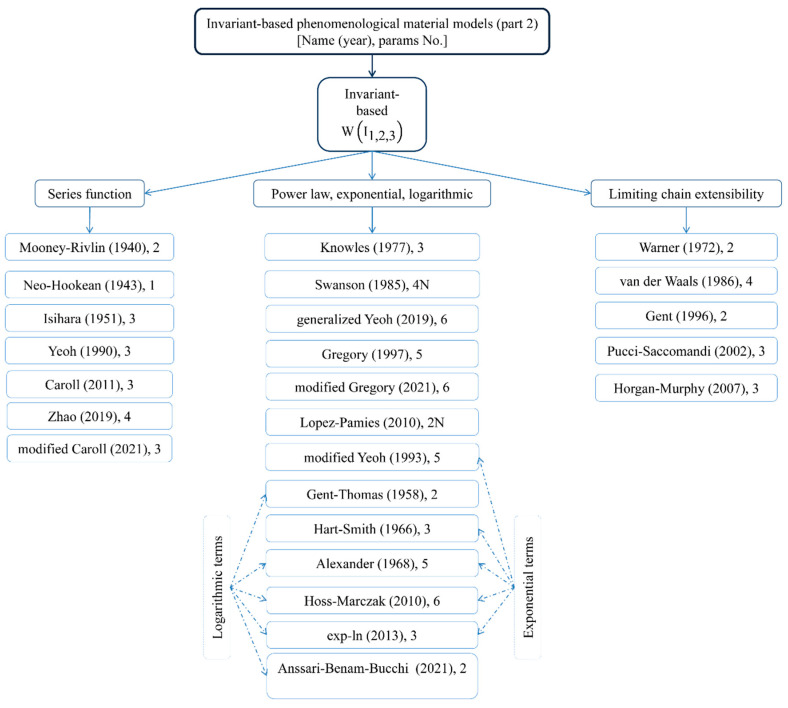
Different phenomenological models of hyper-elastic materials (part 2).

**Figure 3 materials-15-08878-f003:**
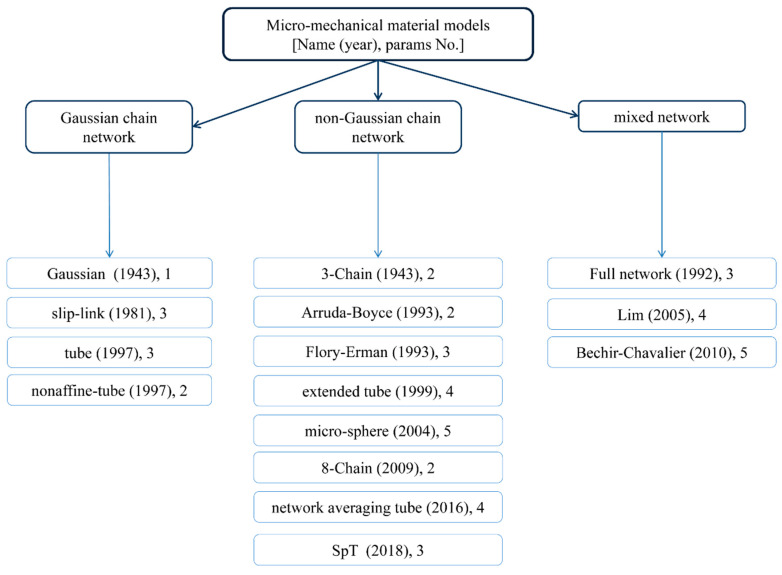
Different micromechanical models of hyper-elastic materials.

**Figure 4 materials-15-08878-f004:**
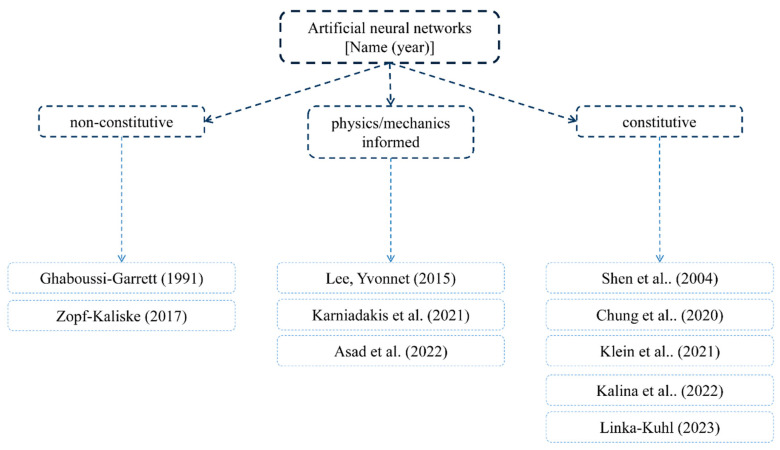
Different ANN (artificial neural networks) models of hyper-elastic materials.

## Data Availability

Not applicable.

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
