# Peer review of "Probabilistic Analysis of Composite Materials with Hyper-Elastic Components"

_materials, 2022, doi:10.3390/ma15248878_

Round 1

Reviewer 1 Report

The probabilistic analysis of composites with components exhibiting hyper-elastic constitutive behavior is reviewed in this manuscript. The theories and methods of the hyper-elastic material with a special emphasis on the recent progress and the trends are provided. The focus is a discussion of up-to-date methods and works in homogenization method and effective medium analysis. It has been oriented on random composites with and without any interface defects.

The reviewer suggested that the article be modified appropriately.

1. As a review article, it is suggested to add relevant graphic illustration and analysis.

2. As shown in Fig.1, the accuracy, practicality and limitations of each model should be clearly stated in this manuscript.

Author Response

Review no 1:

The probabilistic analysis of composites with components exhibiting hyper-elastic constitutive behavior is reviewed in this manuscript. The theories and methods of the hyper-elastic material with a special emphasis on the recent progress and the trends are provided. The focus is a discussion of up-to-date methods and works in homogenization method and effective medium analysis. It has been oriented on random composites with and without any interface defects.

The reviewer suggested that the article be modified appropriately.

[1] As a review article, it is suggested to add relevant graphic illustration and analysis.

Answer: Analysis of the recalled constitutive models have been provided, but according to the fact that after modifications the work includes 51 pages (and 40 various constitutive models) it is impossible to attach any other new elements. Its remarkable extension reflects the Reviewers’ comments.

[2] As shown in Fig.1, the accuracy, practicality and limitations of each model should be clearly stated in this manuscript.

Answer: Figure 1 has been extended to 4 figures and all constitutive models have their own equations, so that now their limitations are more clear.

Reviewer 2 Report

This paper is a review of the literature on probabilistic methods related to composite materials with components exhibiting hyperelastic constitutive behavior. The proposed review and related discussion begins with a general overview of the theories and methods of hyperelastic material. In addition, a review of current trends in probabilistic methods is presented, followed by a review of the literature on interfacial theoretical, experimental, and numerical processing of composites.

However, in the paper, the methods are given superficially, without details. Out of 267 sources, only 7 articles are 2020-2021. This may indicate that the article does not sufficiently disclose the latest trends in this area.

In addition, references to sources are missing in the text: lines 106-107, 112-114.

Author Response

Review no 2:

This paper is a review of the literature on probabilistic methods related to composite materials with components exhibiting hyperelastic constitutive behavior. The proposed review and related discussion begins with a general overview of the theories and methods of hyperelastic material. In addition, a review of current trends in probabilistic methods is presented, followed by a review of the literature on interfacial theoretical, experimental, and numerical processing of composites.

However, in the paper, the methods are given superficially, without details. Out of 267 sources, only 7 articles are 2020-2021. This may indicate that the article does not sufficiently disclose the latest trends in this area. In addition, references to sources are missing in the text: lines 106-107, 112-114.

Answer: The Authors would like to thank the Reviewer very much for this important comment – 43 new references have been recalled in the resubmission, where 9 have been published in 2022, while 7 – in 2021.

Round 2

Reviewer 2 Report

The authors took into account all the comments and suggestions.